# Impact of yoga on cardiometabolic health in adults with overweight or obesity: A systematic review and meta-analysis of randomized controlled trials

Widya Wasityastuti[1☉], Miranti Dewi Pramaningtyas[2☉], Rakhmat Ari Wibowo[1,3]*, Muhammad Luthfi Adnan[4], Rafik Prabowo[4], Zulfa Tsurayya[5], Andika Dhamarjati[5], Justinus Putranto Agung Nugroho[6], Ni Komang Ayu Swanitri Wangiyana[7], Om Lata Bhagat[8], Mumtaz Maulana Hidayat[9], Sameer Badri Al-Mhanna[10,11,12], Vega Pratiwi Putri[13,14], Abdullah F. Alghannam[15], Alexios Batrakoulis[16,17]

1 Department of Physiology, Faculty of Medicine, Public Health, and Nursing, Universitas Gadjah Mada, Yogyakarta, Indonesia, 2 Department of Physiology, Faculty of Medicine, Universitas Islam Indonesia, Sleman, Yogyakarta, Indonesia, 3 Physical Activity for Health Research Centre, University of Edinburgh, Edinburgh, United Kingdom, 4 Faculty of Medicine, Universitas Islam Indonesia, Sleman, Yogyakarta, Indonesia, 5 Faculty of Medicine, Public Health, and Nursing, Universitas Gadjah Mada, Yogyakarta, Indonesia, 6 Department of Physiology, Faculty of Medicine, Duta Wacana Christian University, Yogyakarta, Indonesia, 7 Department of Physiology, Faculty of Medicine, University of Mataram, Mataram, Indonesia, 8 Department of Physiology, All India Institute of Medical Sciences Jodhpur, Jodhpur, India, 9 Department of Physiology, Faculty of Medicine, Universitas Pancasakti, Tegal, Indonesia, 10 Department of Exercise Physiology, School of Medical Sciences, Universiti Sains Malaysia, Kubang Kerian, Kelantan, Malaysia, 11 Center for Global Health Research, Saveetha Medical College and Hospitals, Saveetha Institute of Medical and Technical Sciences, Chennai, India, 12 Department of Higher Studies, Al-Qasim Green University, Babylon, Iraq, 13 Department of Neurology, Faculty of Medicine, Public Health, and Nursing, Universitas Gadjah Mada, Yogyakarta, Indonesia, 14 Centre for Clinical Brain Sciences (CCBS), University of Edinburgh, Edinburgh, United Kingdom, 15 Lifestyle and Health Research Center, Health Sciences Research Center, Princess Nourah bint Abdulrahman University, Riyadh, Saudi Arabia, 16 Department of Life Sciences, European University Cyprus, Nicosia, Cyprus, 17 Department of Physical Education and Sport Science, Democritus University of Thrace, Komotini, Greece

☉ These authors contributed equally to this work.

* r.wibowo@ed.ac.uk

## Abstract

This systematic review examined the effects of yoga interventions on various cardiometabolic health outcomes in adults with overweight or obesity. Seven major electronic databases and two clinical trial databases were searched from inception to November 2024. The search strategy combined keywords related to yoga, blood pressure, lipids, glucose, redox, and inflammation. Two authors independently screened the articles to identify randomized controlled trials that compared yoga alone with either an inactive control group or other types of physical activity interventions among adults with overweight or obesity. Outcome changes were analyzed using a random-effects meta-analysis model. We assessed the risk of bias in individual studies using the Risk of Bias 2 tool and evaluated the quality of evidence for each outcome using GRADEpro. We identified 30 randomized controlled trials comprising a total of 2,689 participants that met our eligibility criteria. Although most

**Data availability statement:** The data that support the findings of this study are within the paper and Supporting information files.

**Funding:** The author(s) received no specific funding for this work.

**Competing interests:** The authors have declared that no competing interests exist.

studies (25/28) did not explicitly recruit individuals with obesity, the mean baseline BMI of participants met our inclusion criteria for overweight or obesity. Twenty-three randomized controlled trials involving 2,313 participants were included in the meta-analyses, which demonstrated that yoga practices likely have substantial beneficial effects on systolic blood pressure (-4.35 mmHg), diastolic blood pressure (-2.06 mmHg) and modest effects below the minimal clinically important differences on lipid profiles (low-density lipoprotein cholesterol -0.08 mmol/L, high-density lipoprotein cholesterol +0.06 mmol/L) with moderate quality evidence. Yoga may also have positive effects on glucose, redox, and inflammation parameters, although the evidence remains uncertain. Ethnic differences and dose–response effects were found in subgroup analyses. Further high-quality studies among Asian populations, as well as additional research in non-Asian populations, are needed to strengthen the evidence base and enhance generalizability. This study provides evidence supporting the inclusion of yoga in clinical guidelines for the treatment of individuals with overweight or obesity. **Protocol registration:** International Platform of Registered Systematic Review and Meta-analysis Protocols (INPLASY) (ID: 2023100068).

## Introduction

A high body mass index (BMI), which includes the categories of overweight and obesity, represents a significant global public health concern, contributing to elevated morbidity and mortality rates associated with various cardiovascular, pulmonary, metabolic, and musculoskeletal diseases [1]. It is estimated that a high BMI was responsible for 5.02 million deaths in 2019 [1]. Accordingly, inactive individuals with overweight or obesity often exhibit metabolic dysregulation and cardiovascular complications due to chronic low-grade inflammation and impaired lipid homeostasis [2]. These populations frequently exhibit physical limitations and diminished functional capacity, which negatively impact musculoskeletal integrity and overall quality of life. In light of these considerations, the prioritization of nonpharmaceutical interventions, such as exercise training aimed at physical fitness, has emerged as a pivotal objective for medical and exercise communities as well as public health policymakers striving to address the negative health outcomes and consequently escalating costs linked to the obesity problem [3–5].

A systematic review demonstrated the safety, feasibility, and efficacy of diverse forms of physical activity (PA) as preventative measures, management strategies, and treatment options for excess body weight [6]. Additionally, PA has been shown to play a crucial role in addressing sarcopenia among individuals with obesity, a condition closely linked to cardiometabolic health complications [7,8]. However, the adoption of an active lifestyle among adults with obesity is lower than that among adults without obesity [9,10]. The bidirectional relationship between overweight and physical activity (PA) is influenced by pain, which results from increased inflammation and the physical burden of excess weight. This pain has emerged as one of the most common barriers to PA among individuals with a high BMI [11]. On the other hand, weight

management and fitness improvement are considered the most common motives for PA among adults with a high BMI [11]. Considering these barriers and motives, it could be argued that low-moderate intensity PA, which is less likely than higher-intensity PA to cause pain, discomfort, or an additional burden from body weight but still improves fitness and aids in weight reduction, may be preferable for individuals with overweight or obesity [12–16].

Yoga is currently a popular exercise modality worldwide [17,18], incorporating physical postures, breathing techniques, meditation, and relaxation poses into a session. This mind-body PA has been documented as a minimal injury, practicable, and cost-effective form of exercise, aiming to induce positive psychophysiological adaptations in populations with excess body weight [19,20]. Yoga could be promoted among adults with a high BMI since this low-moderate-intensity PA could result in improvement of fitness and reduction of weight among people with low fitness baselines [21–23]. In addition, several potential benefits of yoga on cardiometabolic outcomes, including blood pressure, lipid profile, glucose homeostasis, and inflammatory and antioxidant status, which could prevent obesity-related morbidity, may provide additional motives to improve health among people with a high BMI [11,24].

Several systematic reviews have demonstrated the benefits of yoga on lipid profiles, blood pressure, blood glucose, and inflammation [25–29]. While all of these studies consistently reported the beneficial effects of yoga on cardiometabolic factors, there was still high heterogeneity even after sensitivity and subgroup analyses. However, none of these studies conducted sensitivity or subgroup analyses considering high BMI as a potential moderator of the effects of PA on cardiometabolic parameters [30–32]. Given that excess weight can lead to impaired metabolism, individuals with a high BMI may experience greater effects from yoga. Therefore, this systematic review and meta-analysis aimed to assess the impact of yoga practice on a wide spectrum of cardiometabolic health outcomes in individuals with overweight or obesity, focusing specifically on BMI-defined populations and synthesizing a broader set of outcomes than prior reviews.

## Materials and methods

We registered this systematic review protocol on the International Platform of Registered Systematic Review and Meta-analysis Protocols (INPLASY) (ID: 2023100068) [33] and reported it according to the Preferred Reporting Items for Systematic Reviews and Meta-Analyses Protocols (PRISMA-P) [34].

### Eligibility criteria

**Participants.** We included the study if the participants were considered to have a high BMI according to the WHO and WHO Asia–Pacific guidelines [35,36], with thresholds of more than or equal to 23 and 25 for Asian and other ethnicities, respectively. If the study did not explicitly mention overweight, high BMI, or obesity in the abstract or title, we found the participants' baseline BMI in the full text if individual participant data was not available. If the study participants also had other diseases that could influence the effect of PA on cardiometabolic factors, including type 2 diabetes mellitus, chronic kidney disease, cancer, coronary artery disease, thyroid disease, and chronic heart failure, we excluded the study.

**Intervention.** We included studies that evaluated yoga practice as a complex intervention combining posture, breathing, and meditation; posture and breathing; or posture and meditation. Studies that investigated the effects of one yoga component were excluded. We included studies that combined yoga with other health interventions if the single effect (yoga+diet vs yoga vs control) or the complementary effect of yoga could be examined (yoga+pharmacotherapy vs pharmacotherapy).

**Comparator.** We included any studies that allowed us to compare the single effect or the complementary effect of yoga to inactive control or other types of physical activity interventions. By assessing yoga's efficacy relative to other physical activity or exercise intervention, this decision could strengthen the current review to inform physical activity type selection. We excluded studies that examined the complementary effect of other health interventions into yoga, such as those comparing the addition of pharmacotherapy on yoga to yoga alone.

**Outcomes.** We included the studies if they examined one of the following outcomes as continuous variables:

- Blood pressure: systolic blood pressure (SBP), diastolic blood pressure (DBP), and mean arterial pressure (MAP).

- Lipid profile: high-density lipoprotein (HDL) cholesterol, low-density lipoprotein (LDL) cholesterol, very-low-density lipoprotein (VLDL) cholesterol, total cholesterol (TC), and triglycerides (TG) [37].

- Glucose homeostasis: random blood glucose (RBG), postprandial blood glucose (PPBG), fasting blood glucose (FBG), glycated hemoglobin (HbA1c), homeostatic model assessment for insulin resistance (HOMA-IR) [38].

- Inflammation: C-reactive protein (CRP), high-sensitivity C-reactive protein (hs-CRP), tumor necrosis factor alpha (TNF-α), interleukin 1 (IL-1), IL-6, IL-10, N-terminal pro-brain natriuretic peptide (NT-ProBNP), leptin, resistin, and omentin [39].

- Pro- and Antioxidants

Pro-oxidants: Thiobarbituric acid-reactive substances, malondialdehyde (MDA), F2-isoprostanes, lipid peroxidation, 3-nitrotyrosine, hydrogen peroxide, sulfhydryloxidized, myeloperoxidase, 1-palmitoyl-2-(5-oxovaleroyl)-sn-glycero-3-phosphorylcholine (POVPC), 1-palmitoyl-2-glutaroyl-sn-glycero-3-phosphorylcholine (PGPC)

Antioxidants: total antioxidant capacity/status, Trolox equivalent antioxidant capacity (TEAC), glutathione peroxidase, catalase, glutathione, ascorbic acid, nitric oxide, and adenosine deaminase [40,41].

### Study Type

Randomized controlled trials (RCTs) investigating the chronic adaptations of yoga interventions were included.

**Information sources and search strategies.** We conducted systematic searches on seven major databases on August 23, 2023 (Scopus, Web of Science, Medline (OVID), Embase, Cochrane Library, SPORTDiscus, and PsycInfo (OVID)) and two clinical trial databases (WHO-IT CRP and clinicaltrials.gov) via Scopus, using search strategies combining free terms and indexed terms consisting of yoga and outcomes (blood pressure, lipid, glucose, or redox or inflammation). The search strategy was adapted from previous systematic reviews [25–29] on the Medline database and then adapted into other databases (S1 Text). We then rerun the database searches on November 3, 2024 and December 25, 2025. We also manually checked reference lists from previous systematic reviews and included studies [25–29]. If several registered trials met our inclusion criteria but had not yet been published, we contacted the corresponding authors to check the availability of the results. No language restrictions were in place during the literature search.

### Data management

After the database searches were conducted, one author imported the results from all the literature searches into Endnote software, removed duplicates using the software, and manually removed any other duplicates. The results were then imported into Rayyan software [42].

### Study selection

It was anticipated that the initial search strategy would yield several thousand citations. Consequently, we adapted the three-tiered approach used in a scoping review framework [43]. The entire number of studies to be screened was divided into four equal groups [43]. In each group, two independent reviewers screened the titles and abstracts to exclude studies that explicitly violated the inclusion criteria. If the reviewers were unsure whether the studies should be included but found no clear reasons to exclude them on the basis of the title and abstract alone, they included the studies in the "maybe" category for full-text screening. After the titles and abstracts of all the studies were screened, the two reviewers screened the full texts to identify the final set of eligible studies. One reviewer manually searched the reference lists of the eligible studies from database searches and previous systematic reviews. A similar two-stage selection process was conducted to

screen references captured from manual searches. Any discrepancies between reviewers at all stages during the selection process were resolved through discussion facilitated by a third reviewer.

## Data extraction

Two reviewers extracted the data from the included studies using a data extraction form developed based on the Consensus on Exercise Reporting Template (CERT) (S1 Data) [44].

## Risk of bias assessment

Two reviewers assessed the risk of bias in all eligible studies using the Cochrane Collaboration tool for assessing the risk of bias in primary RCTs [45]. Discussions with a third reviewer were conducted to resolve any discrepancies. The authors presented a "risk of bias" table for each study. We assessed the primary RCT studies based on the quality of the randomization process, the effect of the assignment to the intervention, the quality of the outcome assessment, the completeness of the data and handling of incomplete data, and the presence of reporting bias [45]. We did not assess the primary RCTs on whether participants and personnel were blinded to their group allocation since this would not be appropriate for a PA intervention study [46]. We chose to assess the risk of bias for assessing the effect of assignment to the intervention rather than the effect of adhering to the intervention since we assessed the effects of yoga practice in a real-world setting in which participants were not controlled to adhere to the protocol. In the quality of outcome measurement domain, we determined that all outcome measurements, except for manual blood pressure measurements, were not influenced by the outcome assessors' awareness of the participants' allocation and knowledge of the intervention. We assessed a study as having a high risk of bias when there is a high risk of bias in at least one domain.

## Synthesis methods

We conducted a vote-counting approach to provide a narrative synthesis as a preliminary analysis by categorizing the results of each outcome into the following five categories: 1) statistically significant positive effects of yoga practices counted as a + 2 score, 2) nonstatistically significant positive effects of yoga practices counted as a + 1 score, 3) statistically significant negative effects of yoga practices counted as a -2 score, 4) nonstatistically significant negative effects of yoga practices counted as a -1 score, and 5) nonsignificant results without effect direction counted as a 0 score [47]. The score summary for each outcome was obtained as an indication of the effectiveness of yoga practices. Considering the methodological heterogeneity across the included studies, we performed random effects model meta-analyses based on change from baseline to calculate the effect size using the mean difference (MD) when the same measurement scale was used across the studies. Otherwise, we calculated the effect size using the standardized mean difference (SMD). When the individual studies did not provide the standard deviation (SD) of the change from the baseline, the value was obtained by calculating it from the baseline SD and posttreatment SD using the available p value of the change from baseline or coefficient correlation from another included study with similar characteristics. We used the WebPlotDigitizer (Automeris LLC, USA) to extract the change from the baseline and its standard deviation (SD) from the graphical data [47]. Two reviewers extracted the values independently; where minor differences occurred, the values were re-checked against the original graphs and a final value was agreed, We interpreted the treatment effect using a threshold for each outcome [48–57], as presented in Table 1. Funnel plots were generated to assess publication bias through visual inspection of funnel plot symmetry, and this was supplemented by Egger's test.

## Sensitivity and subgroup analyses

We performed sensitivity analyses by excluding studies that had a high risk of bias. Subgroup analyses were conducted to examine the influence of different doses of yoga and participants' ethnicities, as previous studies have reported ethnic

**Table 1. Minimal clinically important differences.**

| Outcome | Threshold |
| --- | --- |
| Blood pressure | |
| Systolic blood pressure | 1 mmHg [48] |
| Diastolic blood pressure | 2 mmHg [49] |
| Lipid profile | |
| LDL | 1 mmol/L [50] |
| HDL | 1 mmol/L [51] |
| Total cholesterol | 1 mmol/L if the baseline of total cholesterol >= 200 mg/dl [52] |
| Glucose metabolism | |
| HbA1c | 0.03% [53] |
| Blood glucose | 0.5 mmol/l [54] |
| Inflammation | |
| CRP | 1 mg/L [55] |
| hs-CRP | 1 mg/L [55] |
| IL-6 | 1 ug/dL [55] |
| Prooxidant and antioxidant | |
| MDA | 5 mmol/mL [56] |
| TEAC | 5.9 mmol/dL [57] |

HDL: high-density lipoprotein cholesterol; LDL: low-density lipoprotein cholesterol; HbA1c: glycated hemoglobin; hs-CRP: high-sensitivity C-reactive protein; IL-6: interleukin-6; MDA: malondialdehyde; TEAC: trolox equivalent antioxidant capacity.

differences in the effects of exercise and varying practices of yoga, with Eastern approaches placing greater emphasis on spirituality than their Western counterparts do [58–60].

## Confidence in cumulative evidence

We assessed the quality of evidence for blood pressure, lipid profile, glucose homeostasis, and inflammatory markers using the Grading of Recommendation, Assessment, Development, and Evaluation (GRADE) approach by checking the risk of bias in the included studies, heterogeneity and its source across the included studies, generalizability of the findings to individuals with excess weight, imprecision of the findings considering confidence intervals and the number of total participants, and publication bias [61].

## Results

Our search across seven databases resulted in a total of 17,024 results (Fig 1). Through an automated deduplication process using the Endnote, 3,100 duplicate records were removed, and a subsequent manual deduplication step removed an additional 1,556 duplicates, leaving 12,368 unique records for screening based on titles and abstracts. Having screened 128 full-text articles for eligibility, we included 17 studies. Manual searches from previous systematic reviews were also conducted to check the eligibility of 116 articles. We found 12 articles meeting the eligibility criteria, but all of them were included through database searches. Updated searches on November 3, 2024 resulted in the addition of 11 studies and December 25, 2025 resulted in the addition of 2 studies. The reasons for exclusion are summarized in S1 Table.

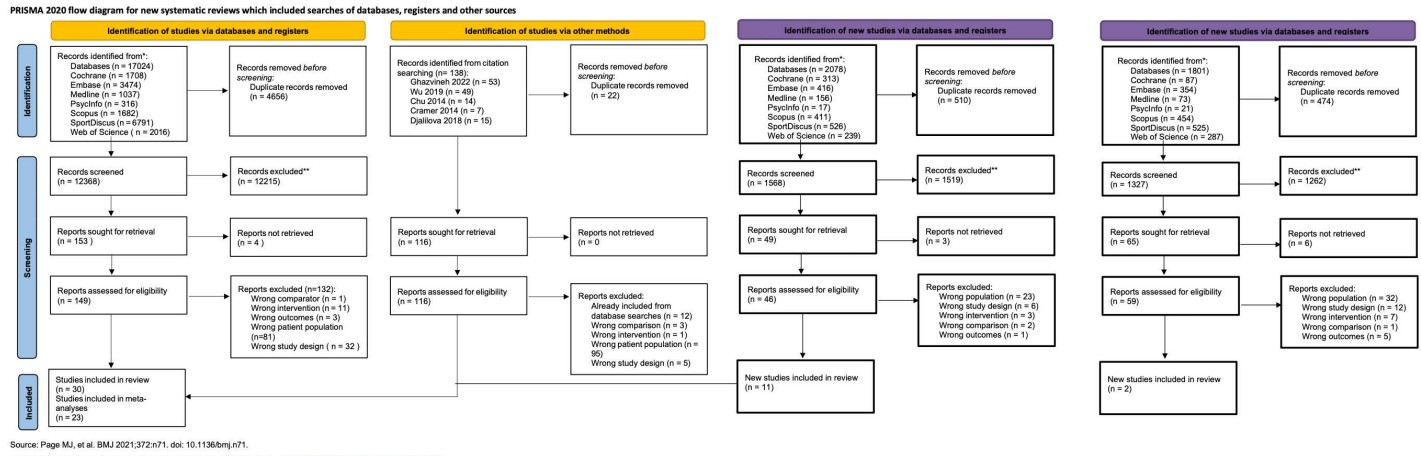

**Fig 1. PRISMA flow chart of study selection.**

## Study characteristics and vote-counting results

The characteristics of the 30 included studies [62–91] are summarized in S2 Table. A total of 23 studies were conducted in Asian countries, with the majority (21 studies) conducted in India, while the remaining studies were conducted in Indonesia and Korea. Among the seven studies conducted outside of Asia, five were carried out in the United States, one in Germany, and one in Australia. The included studies included a total of 2,689 participants, with sample sizes ranging from 8--383 in the yoga intervention group and from 8--375 in the control group. Most studies included both male and female participants, with the proportion of male participants ranging from 9% to 77.8%. However, six studies recruited only female participants, and three studies exclusively recruited male participants. Only three studies specifically targeted individuals with overweight or obesity in their recruitment. Twenty studies reported blood pressure outcomes, with 15 focusing on individuals with prehypertension, primarily because of high systolic blood pressure (SBP). Among the 12 studies examining fasting blood glucose (FBG), 5 studies included participants with impaired FBG, and 1 study included participants with impaired HbA1c. Five studies reporting postprandial blood glucose (PPBG) outcomes were conducted among the Asian population; 2 of these studies included participants with impaired PPBG, and 1 study included participants with impaired HbA1c. Four studies examined HOMA-IR, with three involving participants with impaired glucose parameters. Three of four studies on HbA1c were conducted among the Asian population, and two of these studies involved participants with impaired blood glucose parameters. Eleven studies examined LDL, with only 1 study including participants with high LDL levels. Among the seven studies reporting HDL outcomes, four included participants with undesirable HDL levels. Two of the eleven studies examining triglyceride (TG) levels included participants with high TG levels. Among the eleven studies reporting total cholesterol (TC) levels, four included participants with high TC levels. Only three studies reported VLDL outcomes, all of which were conducted among Asian populations.

The available studies examining redox status include markers such as MDA, homocysteine, glutathione (GSH), vitamin E, vitamin C, superoxide dismutase (SOD), and catalase. At least two studies were available for each marker, except for catalase and homocysteine. Studies have assessed the levels of the inflammatory markers, including hs-CRP, TNF-α, IL-1, IL-6, IL-10, soluble interleukin-2 receptor (sIL-2R), and adiponectin. Adiponectin and sIL-2R levels were reported by only one study for each outcome.

Based on vote counting, compared with inactive controls, yoga practices might have more beneficial effects on blood pressure (SBP -27 scores, DBP -18 scores), lipid profiles (LDL -9 scores, TG -9 scores, TC -8 scores, VLDL -3 scores,

PLOS Global Public Health

HDL + 7 scores), most blood glucose parameters (FBG -9 scores, PPBG -4 scores, HOMA-IR -7 scores), pro-oxidant markers (MDA -3 scores, homocysteine -2 scores), antioxidant markers (GSH + 5 scores, SOD + 4 scores, vitamin C + 3 scores, vitamin E + 1 scores, catalase +2 scores), pro-inflammatory markers (TNF-α -8 scores, IL-1–4 scores, IL-6–4 scores, hs-CRP -2 scores, sIL-2R -2 scores), and anti-inflammatory markers (IL-10 + 4 scores, adiponectin +2 scores). However, vote counting on HbA1c showed a detrimental effect (+1 point). Compared with walking exercise, vote counting results favored yoga for lipid profile and glucose parameter outcomes but conflicting results for blood pressure outcomes. In addition, vote counting results found similar effect of yoga compared to moderate intensity aerobic exercise on blood pressure (S3 Table).

### Risk of bias analysis

We assessed the risk of bias in twenty-two studies consisting of 2313 participants eligible for meta-analyses. In general, most studies raised concerns about bias. Studies with a high risk of bias were particularly affected by a lack of information on their randomization processes and baseline imbalances, which could introduce selection bias. Additionally, missing outcome data raised concerns about attrition bias, whereas bias in the measurement of blood pressure could lead to detection bias. Most studies did not provide their protocols, raising concerns in domain 5 (S4 Table).

### Meta-analyses

**Glucose metabolism.** Overall, meta-analyses showed yoga practices benefit glucose metabolism among individuals with overweight or obesity, particularly glucose metabolism outcomes measured via precise measurement tools, such as HbA1c and HOMA-IR. While yoga practices showed benefits on HbA1c (mean difference -0.04%, 95% confidence interval [CI]: -0.07% to -0.02%, $I^2 = 54\%$) and HOMA-IR (mean difference -0.87 95% CI: -1.64 to -0.10, $I^2 = 94\%$), yoga practices did not provide benefits for less precise measurements of glucose metabolism, such as fasting blood glucose (mean difference -0.11 mmol/L, 95% CI: -0.26 to 0.03, $I^2 = 90\%$) and postprandial blood glucose (mean difference -0.20 mmol/L, 95% CI: -0.62 to 0.23 mmol/L, $I^2 = 96\%$) (Fig 2). Subgroup analysis by ethnicity further confirmed that yoga interventions did not significantly impact fasting blood glucose levels in individuals with high BMIs, regardless of ethnic background. However, considerable heterogeneity persisted in both the Asian and non-Asian populations. The frequency of yoga sessions may account for some of this heterogeneity within the non-Asian group, as Hunter (2018) provided sessions at least three times per week, whereas Yang (2011) offered only two sessions per week.

**Lipid profile.** Meta-analyses of yoga intervention studies generally showed beneficial effects of yoga on lipid profiles, including reductions in VLDL (-0.04 mmol/L, 95% confidence interval [CI]: -0.08 to -0.00, $I^2 = 0\%$) and triglycerides (-0.26 mmol/L, 95% CI: -0.42 to -0.11, $I^2 = 86\%$), as well as improvements in HDL (+0.06 mmol/L, 95% CI: 0.01 to 0.10, $I^2 = 73\%$) (Fig 3). However, the effects varied by ethnicity. Yoga improved the lipid profile in the Asian population, whereas no statistically significant effects were detected in the non-Asian population. In contrast, meta-analyses revealed no effect of yoga on total cholesterol (-0.06 mmol/L, 95% CI: -0.16 to 0.03, $I^2 = 81\%$) and LDL (-0.08 mmol/L, 95% CI: -0.19 to 0.04, $I^2 = 83\%$), with the effect direction differing between Asian and non-Asian subjects.

**Blood pressure.** Overall, meta-analyses of 16 studies involving 962 subjects with a high BMI showed that yoga reduced systolic and diastolic blood pressure, with substantial heterogeneity: systolic blood pressure decreased by -4.35 mmHg (95% confidence interval [CI]: [-6.10 to -2.60, $I^2 = 83\%$) and diastolic blood pressure decreased by -2.06 mmHg (95% CI: -2.97 to -1.15, $I^2 = 69\%$) (Fig 4). Subgroup analyses suggested that ethnicity could be a source of this heterogeneity. Among Asian participants, yoga interventions significantly reduced systolic blood pressure by -5.52 mmHg (95% CI: -7.24 to -3.80 mmHg) and diastolic blood pressure by -2.81 mmHg (95% CI: -3.86 to -1.76 mmHg), still with substantial heterogeneity. In contrast, no statistically significant reduction in blood pressure was observed in the non-Asian population.

**(a) Fasting blood glucose**

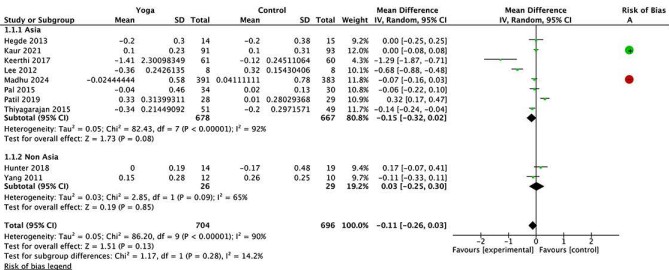

**(c) Hemoglobin A1C**

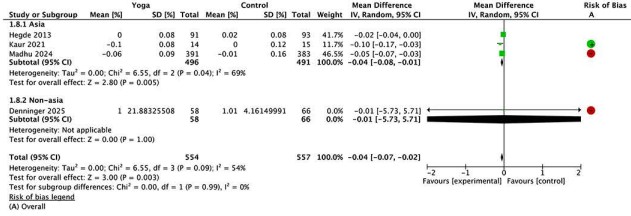

**(b) Post-prandial blood glucose**

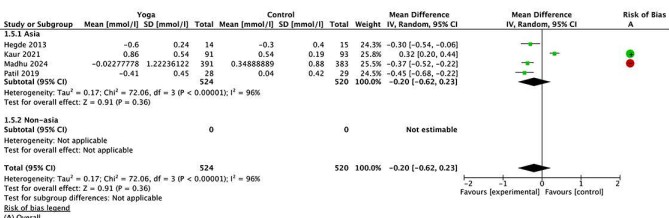

**(d) Homeostatic Model Assessment for Insulin Resistance**

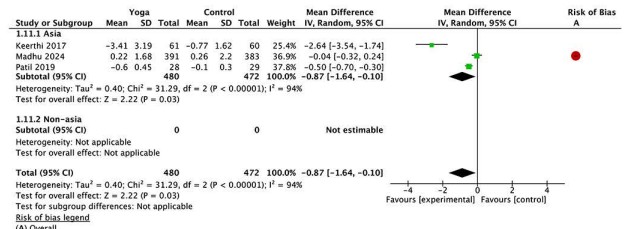

**Fig 2. Meta-analyses and subgroup analyses by ethnicity of the effect of yoga practices on glucose metabolism.**

**Redox profile.** Three studies examined the effects of yoga practices on pro-oxidant markers: one study on homocysteine and two studies on MDA. However, none of these studies provided sufficient data for further meta-analysis. Regarding antioxidant parameters, studies involving only Asian subjects showed statistically significant beneficial effects of yoga on glutathione, with a standardized mean difference (SMD) of 5.89 (95% confidence interval [CI]: 2.09 to 9.70, $I^2 = 98\%$). No statistically significant effects were observed for vitamin C, vitamin E, or SOD (Fig 5).

**Inflammation profile.** Five inflammation markers and one anti-inflammatory marker were included in the meta-analyses. Pooled effects showed that yoga practices resulted in statistically significant reductions in most inflammation markers: TNF-alpha (-1.46, 95% confidence interval [CI]: -1.93 to -0.99, $I^2 = 30\%$), IL-1 (-0.48, 95% CI: -0.74 to -0.21, $I^2 = 43\%$), IL-6 (-0.63, 95% CI: -0.87 to -0.39, $I^2 = 0\%$), and hs-CRP (-0.38, 95% CI: -0.68 to -0.07, $I^2 = 38\%$), as well as an improvement in the anti-inflammatory marker IL-10 (0.39, 95% CI: 0.15 to 0.63, $I^2 = 22\%$) (Fig 6). Ethnic differences could not be statistically explored as a source of heterogeneity in the inflammation markers because only one study among non-Asian subjects examined the hs-CRP outcome and one study among non-Asian subjects examined IL-6.

### Subgroup and sensitivity analyses

We conducted subgroup and sensitivity analyses to examine the impact of intervention dosage and study quality on heterogeneity (S5 Table). However, subgroup analyses based on participants' baseline health status (e.g., high blood glucose below the threshold for type 2 diabetes, high blood pressure below the threshold for hypertension, and impaired lipid status below the threshold for dyslipidemia or metabolic syndrome) were not possible due to the limited number of included studies. In addition, subgroup analysis based on yoga intensity was not possible because all included studies did not provide information. Our subgroup analyses generally indicated that intervention duration, session duration, session frequency, and study quality influenced the effects of yoga on several cardiometabolic outcomes.

## (a) Low-density lipoprotein cholesterol

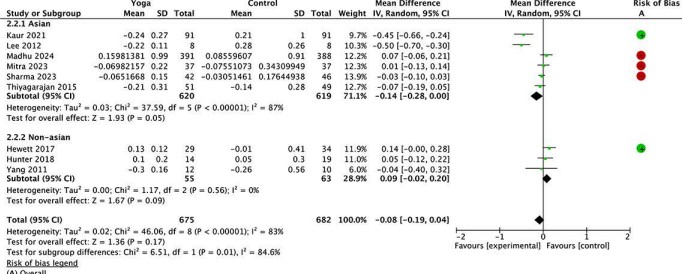

## (b) High-density lipoprotein cholesterol

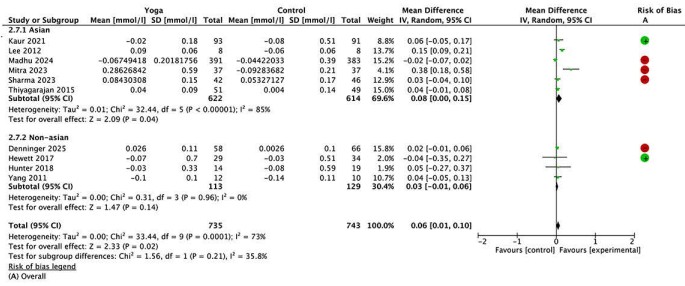

## (c) Triglycerides

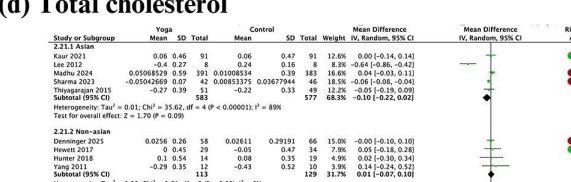

## (d) Total cholesterol

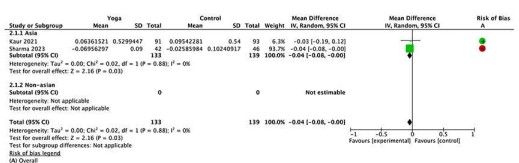

## (e) Very-low-density lipoprotein cholesterol

**Fig 3. Meta-analyses and subgroup analyses by ethnicity of the effect of yoga practices on lipid profile.**

Studies with a low risk of bias demonstrated greater effects on cardiometabolic outcomes when the intervention lasted at least 12 weeks, with sessions lasting at least 60 minutes and conducted at least three times per week. Subgroups with these characteristics showed beneficial effects of yoga with minimal to moderate heterogeneity on LDL (-0.48 mmol/L, 95% confidence interval [CI]: -0.62 to -0.33), HDL (+0.12 mmol/L, 95% CI: +0.03 to +0.20), triglycerides (-0.35 mmol/L, 95% CI: -0.47 to -0.23), systolic blood pressure (SBP) (-8.96 mmHg, 95% CI: -11.73 to -6.20 mmHg), and diastolic blood pressure (DBP) (-9.09 mmHg, 95% CI: -11.88 to -6.30 mmHg) among the Asian population. In contrast, yoga with these dosage characteristics did not influence fasting blood glucose, postprandial blood glucose, or HbA1c in the Asian population. However, substantial heterogeneity remained in postprandial blood glucose and HbA1c, which might be explained by differences in participants' baseline blood glucose levels. Due to the limited number of studies examining yoga interventions with durations shorter than 12 weeks and session lengths under 60 minutes, we were unable to perform both subgroup and sensitivity analyses, particularly in terms of excluding studies with a high risk of bias.

### Publication bias

Since the funnel plots were symmetric (Fig 7) and Egger's tests were non-significant (all p ≥ 0.05), there was no evidence of strong publication bias on blood pressure and blood glucose outcomes. Because of the limited number of studies, publication bias could not be assessed for other outcomes.

### Quality of evidence

The meta-analyses of all the outcomes included randomized controlled trials (RCTs), some of which had a high risk of bias (S6 Table). Substantial heterogeneity was observed in the meta-analysis results for these outcomes, which could

# (a) Systolic blood pressure

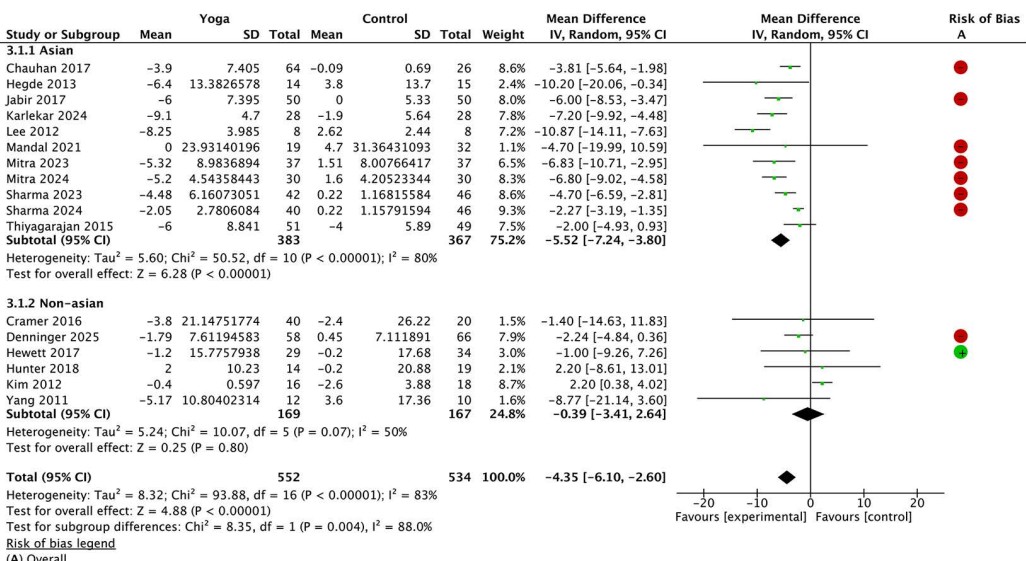

# (b) Diastolic blood pressure

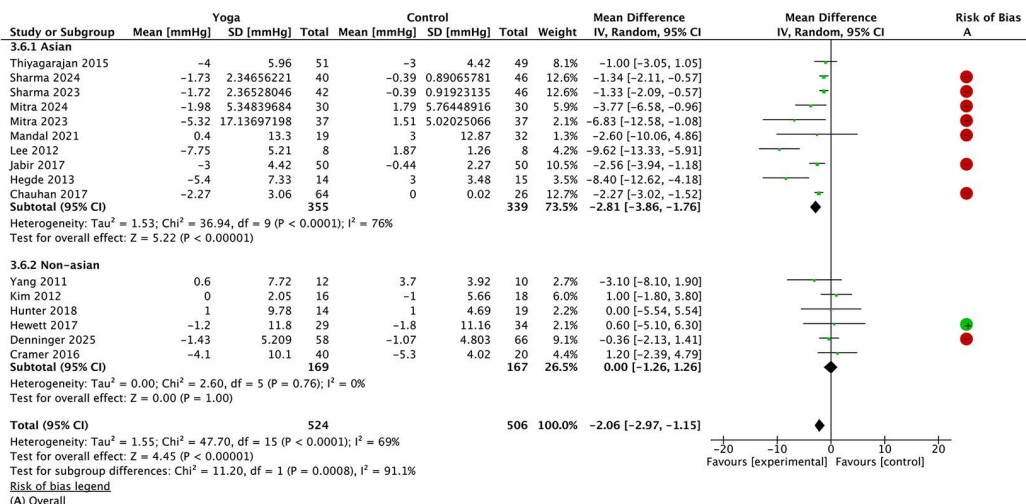

**Fig 4. Meta-analyses and subgroup analyses by ethnicity of the effect of yoga practices on blood pressure.**

be partly explained by ethnic differences and dose–response effects, except for HOMA-IR and HbA1c. For most blood glucose, lipid profile, and blood pressure outcomes, the results did not surpass the minimal clinically important difference (MCID) and were based on a pooled sample of more than 400 participants, except for postprandial blood glucose (PPBG), HbA1c, diastolic blood pressure (DBP), and VLDL. Consequently, the quality of evidence for most blood glucose, lipid profile, and blood pressure outcomes was downgraded by one level to moderate quality. For HbA1c, HOMA-IR, VLDL, and DBP, the quality of evidence was downgraded by two to three levels, resulting in very low to low quality evidence.

# (a) Vitamin C

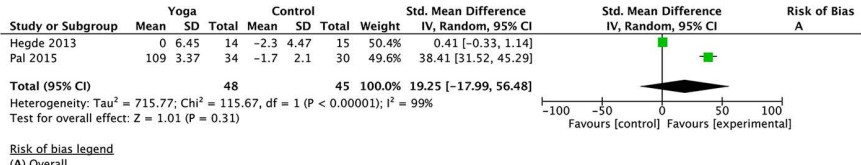

**Risk of bias legend**
(A) Overall

# (b) Vitamin E

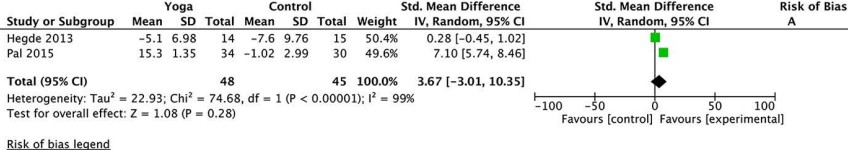

**Risk of bias legend**
(A) Overall

# (c) Superoxide dismutase

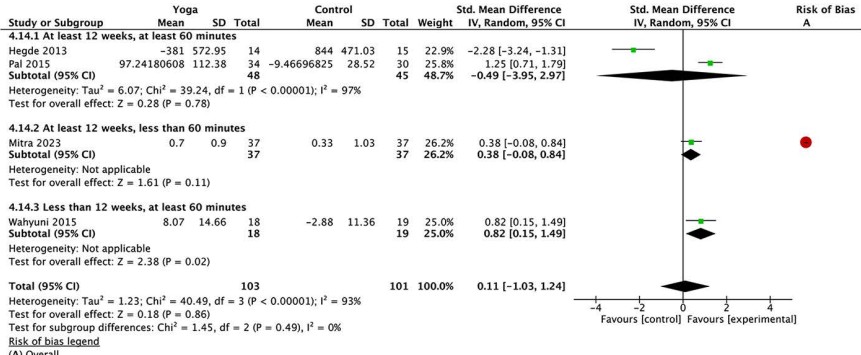

# (d) Gluthatione

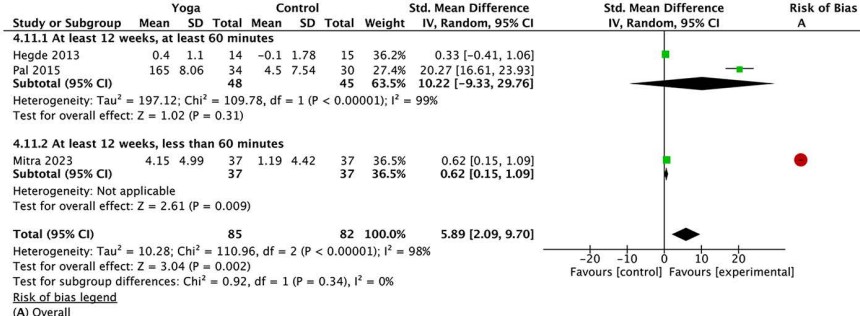

**Fig 5. Meta-analyses and subgroup analyses by ethnicity of the effect of yoga practices on redox profile.**

In addition to the inclusion of studies with a high risk of bias, most meta-analyses on antioxidant, proinflammatory, and anti-inflammatory markers showed unexplained heterogeneity and were based on fewer than 400 participants. As a result, the quality of evidence for these outcomes was downgraded by two to three levels, leading to low to very low-quality

## (a) Interleukin-1

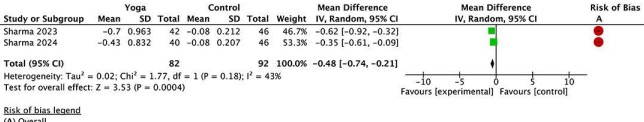

## (b) Interleukin-6

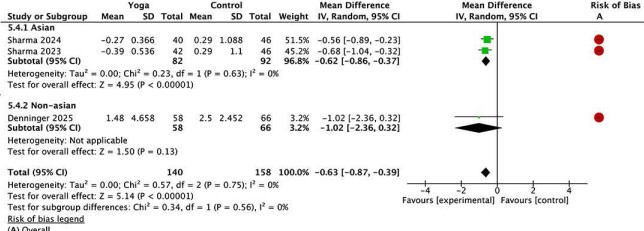

## (c) Interleukin-10

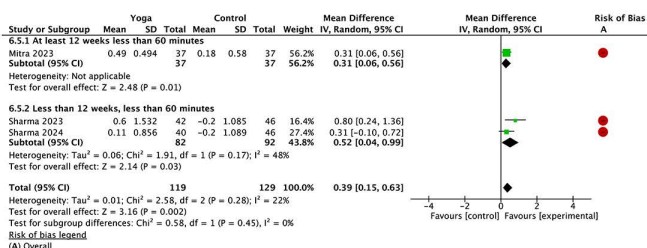

## (d) Tumour Necrosis Factor alpha

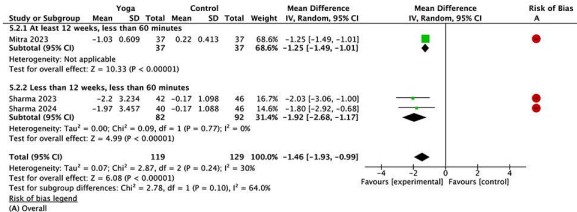

## (e) High-sensitivity C-reactive protein

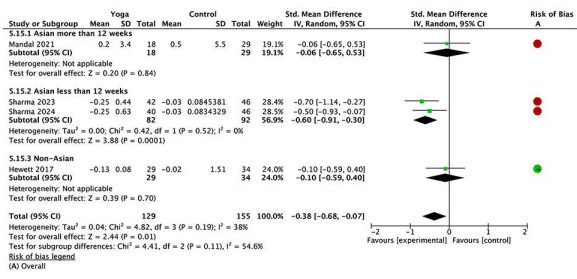

**Fig 6. Meta-analyses and subgroup analyses by ethnicity of the effect of yoga practices on inflammation profile.**

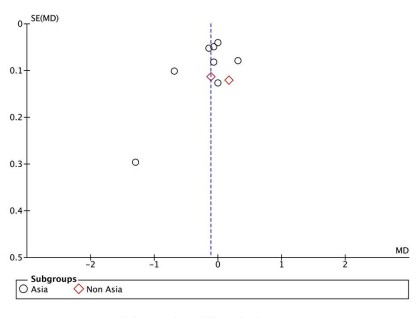

(a) Fasting blood glucose

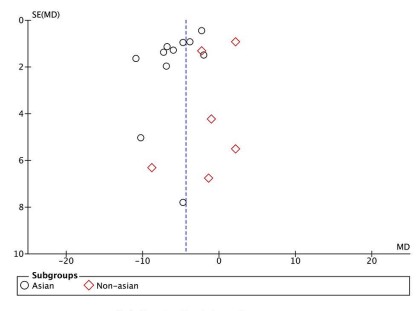

(b) Systolic blood pressure

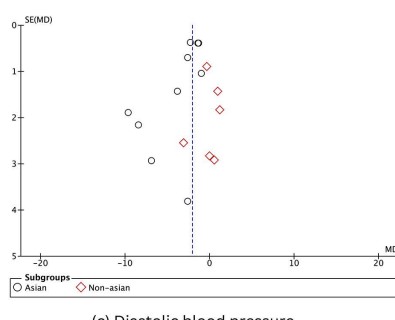

(c) Diastolic blood pressure

**Fig 7. Funnel plots of publication bias.**

evidence. For most outcomes, unexplained heterogeneity was due to the limited number of primary studies available to conduct subgroup analyses for non-Asian subjects, as well as for yoga practices with less than 60 minutes per session, fewer than three sessions per week, or a duration of less than 12 weeks.

In the ethnicity-based subgroup analyses of lipid profiles and blood pressure outcomes, the quality of evidence for most outcomes among the non-Asian subgroup remained moderate, with one upgrade due to the absence of studies with a high risk of bias and one downgrade due to the small number of participants (fewer than 400). Among the limited number of studies involving non-Asian subjects on blood glucose outcomes, we were only able to assess the pooled effect

on fasting blood glucose (FBG), which showed substantial heterogeneity and warrants further investigation, resulting in low-quality evidence.

Following sensitivity analysis by excluding studies with a high risk of bias among Asian subjects and conducting subgroup analyses based on the dose of yoga practices, the quality of evidence was generally downgraded by one additional level due to the limited number of studies available, resulting in imprecision and unexplained heterogeneity. This low-quality evidence was primarily observed in studies with yoga interventions that had durations shorter than 12 weeks, session lengths under 60 minutes, or frequencies of fewer than three times per week. In contrast, studies with yoga practices lasting at least 12 weeks, with sessions lasting at least 60 minutes and a frequency of at least three times per week, generally provided moderate-quality evidence.

## Discussion

The present results indicate that yoga promotes beneficial alterations in particular cardiometabolic health-related indicators, such as blood pressure and lipid profiles, among individuals with overweight or obesity. Yoga practices resulted in slight reductions in LDL and triglycerides, a large reduction in SBP and a slight improvement in HDL. Yoga practices may result in little difference in FBG, PPBG, HbA1c, HOMA-IR, DBP, and various antioxidants, as well as proinflammatory and anti-inflammatory markers; however, the evidence remains uncertain. Therefore, our study provides additional evidence for the recent ACSM consensus on the role of PA and exercise in individuals with excess weight, suggesting that yoga, as a mind-body form of PA [92], could not only increase well-being but also offer benefits for cardiometabolic health. In addition, our findings, derived from a large number of studies conducted in Asian countries, provide evidence-based support for the benefits of yoga among populations where it is most commonly practiced [93]. Our findings also provide preliminary evidence that yoga attracted interest from individuals with overweight or obesity, as 25 of our included studies, despite not specifically recruiting this population, had participants with an average BMI in the overweight or obese range.

Our meta-analyses found ethnic differences in the effects of yoga interventions on cardiometabolic factors among individuals with a high BMI, with Asian participants experiencing more favorable outcomes than non-Asian participants. These differences may be attributed to variations in physiological factors across ethnicities and differences in yoga practices across countries [58–60,94]. Yoga practices among Asian participants may involve greater intensity and a greater emphasis on the spiritual component, potentially contributing to the more favorable outcomes observed in this group [60,94,95]. Our results strengthened findings from observational studies suggesting different effects of similar amounts of PA on blood pressure, lipid profiles, and glucose metabolism between ethnicities [58,59]. While a previous systematic review suggested that PA or exercise has a similar effect on visceral adipose tissue, it could be suggested that physiological factors other than visceral adipose tissue moderate ethnic differences in the effects of yoga and other types of exercise on cardiometabolic health [96]. In addition, different metabolic intensities of yoga practices across countries should also be explored in future studies as an alternative cause of the ethnic differences in the effects of yoga practices [21].

The dose–response effects of yoga on cardiometabolic indicators were also found to favor yoga practice at least three times a week with a 60-minute duration in each session for at least 12 weeks. The total minimum duration of yoga practices per week to provide the most beneficial effects on cardiometabolic factors, which was equivalent to at least 180 minutes per week, was greater than the minimum recommendation of 150 minutes of moderate-vigorous PA since yoga practice is considered a low-moderate intensity PA requiring a longer duration to elicit comparable energy expenditure to moderate-vigorous PA [97,98]. Our evidence on the dose–response effect of yoga practice as a low-intensity PA on cardiometabolic factors could serve as additional evidence for future guidelines since the current guidelines do not yet find sufficient evidence on specific doses of low-intensity PA in treating individuals with excess body weight [92]. Yoga practices less than 180 minutes per week might result in beneficial effects lower than the reference, but the evidence is very uncertain because of the limited number of RCTs with a small number of total participants, and some of them have a high risk of bias, requiring further high-quality studies examining the low volume of yoga practices. Further studies with robust

randomization processes and more transparent reporting are needed to provide stronger evidence of low-volume yoga practice effects.

## Glucose metabolism

Our findings indicated that yoga practice might result in small beneficial effects on both metabolism and clinical markers of glucose, as demonstrated by its effect on HOMA-IR, FBG (-0.11 mmol/L), PPBG and HbA1c in people with excess body weight. These subclinical effects may be attributed to the baseline glucose metabolism status of the participants in the included studies, most of whom had normal fasting and postprandial blood glucose levels as well as normal glycated hemoglobin. However, these effects were greater than the effects of yoga on glucose parameters among a mixture of individuals with normal and excess weight (-0.0036 mmol/L FBG) [99] but smaller than those among individuals with type 2 diabetes mellitus (-2.12 mmol/L FBG) [100]. These graded effects might indicate that yoga had varied effects across the baseline glucose metabolism level, with greater effects among individuals with greater impaired glucose metabolism. Therefore, yoga could still be considered beneficial to people with overweight and obesity since it could improve insulin sensitivity, which is usually impaired in this population [101]. Our findings, along with those of a previous network meta-analysis, provided comprehensive evidence of the different effect sizes of exercise modalities on glucose management among individuals with excess weight; yoga had a smaller effect than did moderate-vigorous endurance exercise (-0.22 mmol/L FBG), resistance training (-0.15 mmol/L FBG), or a combination of endurance and resistance training (-0.26 mmol/L FBG) [102].

## Lipid metabolism

In our meta-analysis, yoga resulted in subclinical beneficial changes in VLDL, HDL, and TG but not in TC or LDL. Importantly, these improvements were observed only in Asians, suggesting that ethnic differences in lipid metabolism align with the findings of a previous systematic review [25]. We also found that the heterogeneity in the previous systematic review could be explained by excess weight and the dose of yoga, indicating that yoga could provide greater beneficial effects on the lipid profile among Asian individuals with excess weight than among general Asian individuals [25]. Excess body weight may adversely affect lipid homeostasis control due to complex, low-grade chronic inflammation [1]. Our findings also provide additional evidence from a recent network meta-analysis on the comparative efficacy of different types of exercise on blood lipids, suggesting that resistance training alone, combined aerobic and resistance exercise, and high-intensity interval training have beneficial effects on lipid metabolism markers in individuals with overweight or obesity [102,103]. Therefore, future research should compare yoga practices with other effective exercise interventions with respect to improvements in blood lipids in this population.

## Blood pressure

We observed substantial improvements in SBP and DBP, -4.35 mmHg and -2.06 mmHg, respectively, indicating that yoga may be an effective exercise mode for enhancing resting cardiovascular function in this population. The effect size of yoga on blood pressure among individuals with overweight and obesity in our study was similar to the effect size reported in a previous systematic review that included both individuals with normal weight and those with excess weight (SBP: −5.2 mmHg; DBP: −2.8 mmHg) [26]. However, the previous review did not consider weight variations, frequency and session duration of yoga practices or ethnicity as potential moderators of the outcomes [26]. Our methodological comprehensiveness, which captured two included studies [75,88] that were missed by Wu et al. (2019) [26], allowed us to conduct more comprehensive analyses. However, the effects of yoga on blood pressure are still inferior to those of combined aerobic and resistance training, which has been reported as the most effective training modality for lowering blood pressure in people with excess body weight without comorbidities (-5.58 mmHg SBP and -4.70 DBP) [102]. Combined training has also been shown to be superior in individuals with excess body weight and cardiometabolic comorbidities [104,105].

Consequently, the present findings underscore the potential of yoga practice in inducing meaningful reductions in blood pressure levels among individuals with overweight or obesity.

### Chronic low-grade inflammation and redox status

The present meta-analysis showed that yoga practices might improve antioxidant and anti-inflammatory effects as well as reduce proinflammatory marker levels, but the evidence is uncertain. The uncertainty of the current evidence was caused by the limited number of studies examining these outcomes, particularly those without a high risk of bias. Therefore, future research is necessary to identify the effectiveness of yoga for improving cardiometabolic health through the modification of inflammation and oxidant markers in individuals with overweight or obesity.

### Future directions

Although current exercise prescription guidelines are available for adults with overweight or obesity and yoga has been reported as a popular exercise solution globally [17,92], further research is warranted to elucidate the specific yoga practice parameters (e.g., frequency, intensity, and time). In addition, future real-world interventions utilizing Yoga practices that consider implementation factors could also support clinicians and practitioners in providing evidence-based yoga practice recommendations for people with excess weight to achieve clinically significant improvements [106].

### Limitations

Although our systematic review was assessed as having a high-quality methodology based on the A MeaSurement Tool to Assess Systematic Reviews (AMSTAR-2) checklist (S1 Checklist) [107] and demonstrated good reporting (S2 Checklist), several limitations were identified in the included studies and in our review processes. Several studies conducted among Asian subjects were found to be at high risk of bias, particularly concerning the randomization process, the handling of missing outcome data, and the measurement of outcomes. Second, the limited number of eligible RCTs recruiting non-Asian participants does not provide a comprehensive understanding of the role of yoga practice across a wide spectrum of cardiometabolic health outcomes. However, our findings may help shape future research agendas to generate evidence on the effects of yoga on these outcomes among individuals with obesity in Western countries, where yoga is widely perceived as offering cardiometabolic benefits [108]. With further research, this public perception can become evidence-based. Third, the exclusion of participants with overweight/obesity having comorbidities from the present study may have resulted in a smaller total sample size. This decision helped to isolate prevention-oriented effects of yoga on cardiometabolic outcomes by reducing confounding from established obesity-related disease. However, it also reduces generalizability to clinical practice, given the high prevalence of comorbidities among individuals with overweight or obesity. On contrary, this decision impacted the generalizability of the current systematic review results to the broader target population in clinical practice as there is high prevalence of comorbidities among individuals with overweight or obesity. Fourth, our systematic review included primary studies that did not specifically recruit individuals with obesity but also included studies where the average BMI of participants exceeded the normal threshold to ensure comprehensive coverage of available research. While this approach may have led to the inclusion of individuals with a normal BMI, potentially influencing the outcomes, we were unable to conduct further analyses to examine this impact because of the limited number of primary studies. Future individual participant data meta-analyses are needed to evaluate intervention effects more precisely among individuals with overweight or obesity. Finally, we were unable to conduct meta-regression to statistically assess the dose-response effects due to the limited number of available primary studies.

### Conclusions

This systematic review and meta-analysis present novel data on the beneficial role of yoga in promoting more favorable cardiometabolic health profiles among adults with overweight or obesity. Our findings indicate that, compared with control conditions, yoga significantly impacts select cardiometabolic health-related factors, including insulin resistance, lipid

metabolism, and blood pressure, in individuals with excess weight. These findings support the use of yoga as an adjunctive treatment option for the clinical management of poor metabolic health with potential implications for future clinical guideline development and public health policy focused on lifestyle-based management of overweight and obesity. Nevertheless, further superior methodological-quality RCTs are necessary to elucidate the dose–response relationship and optimal training parameters underlying these encouraging alterations. This information will assist clinicians and practitioners in prescribing yoga interventions to individuals with overweight or obesity.

## Supporting information

**S1 Text. Search strategy.**
(PDF)

**S1 Data. Data extraction form.**
(XLSX)

**S1 Table. List of excluded studies.**
(DOCX)

**S2 Table. Characteristics of included studies.**
(DOCX)

**S3 Table. Vote counting.**
(DOCX)

**S4 Table. Risk of bias analysis.**
(DOCX)

**S5 Table. Subgroup and sensitivity analyses.**
(PDF)

**S6 Table. GRADE analyses.**
(DOCX)

**S1 Checklist. AMSTAR-2 checklist.**
(PDF)

**S2 Checklist. PRISMA checklist.**
(PDF)

## Author contributions

**Conceptualization:** Widya Wasityastuti, Miranti Dewi Pramaningtyas, Rakhmat Ari Wibowo, Muhammad Luthfi Adnan, Mumtaz Maulana Hidayat, Alexios Batrakoulis.

**Data curation:** Miranti Dewi Pramaningtyas, Rakhmat Ari Wibowo, Muhammad Luthfi Adnan, Rafik Prabowo, Zulfa Tsurayya, Andika Dhamarjati, Justinus Putranto Agung Nugroho, Ni Komang Ayu Swanitri Wangiyana, Mumtaz Maulana Hidayat.

**Formal analysis:** Rakhmat Ari Wibowo, Muhammad Luthfi Adnan, Andika Dhamarjati, Vega Pratiwi Putri.

**Funding acquisition:** Widya Wasityastuti, Abdullah F. Alghannam.

**Methodology:** Miranti Dewi Pramaningtyas, Rakhmat Ari Wibowo, Om Lata Bhagat, Sameer Badri Al-Mhanna, Alexios Batrakoulis.

**Supervision:** Widya Wasityastuti, Miranti Dewi Pramaningtyas, Rakhmat Ari Wibowo, Alexios Batrakoulis.

**Validation:** Widya Wasityastuti, Miranti Dewi Pramaningtyas, Rakhmat Ari Wibowo.

**Visualization:** Rakhmat Ari Wibowo.

**Writing – original draft:** Widya Wasityastuti, Miranti Dewi Pramaningtyas, Rakhmat Ari Wibowo, Muhammad Luthfi Adnan, Alexios Batrakoulis.

**Writing – review & editing:** Widya Wasityastuti, Miranti Dewi Pramaningtyas, Rakhmat Ari Wibowo, Muhammad Luthfi Adnan, Rafik Prabowo, Zulfa Tsurayya, Andika Dhamarjati, Justinus Putranto Agung Nugroho, Ni Komang Ayu Swanitri Wangiyana, Om Lata Bhagat, Mumtaz Maulana Hidayat, Sameer Badri Al-Mhanna, Vega Pratiwi Putri, Abdullah F. Alghannam, Alexios Batrakoulis.

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
