## [Decision Letter · Decision Letter 0]

23 Dec 2025

PGPH-D-25-01226

Impact of yoga on cardiometabolic health in adults with overweight or obesity: A systematic review and meta-analysis of randomized controlled trials

Dear Dr. Batrakoulis,

Thank you for submitting your manuscript to PLOS Global Public Health. After careful consideration, we feel that it has merit but does not fully meet PLOS Global Public Health’s publication criteria as it currently stands. Therefore, we invite you to submit a revised version of the manuscript that addresses the points raised during the review process.

EDITOR: Dear Author, please attend to all the comments provided by the reviewers and make necessary corrections.

We look forward to receiving your revised manuscript.

Kind regards,

Zulkarnain Jaafar

Academic Editor

Journal Requirements:

2. We note that you have included your Figures within the body of your manuscript. Please remove the Figures from the body of your manuscript and upload them as separate Figure files.

3. As required by our policy on Data Availability, please ensure your manuscript or supplementary information includes the following:

4. In the online submission form, you indicated that “The data that support the findings of this study are available from the corresponding author upon reasonable request.”.

3. Uploaded as supplementary information.

Additional Editor Comments (if provided):

Reviewers' comments:

Reviewer's Responses to Questions

**Comments to the Author**

1. Does this manuscript meet PLOS Global Public Health’s publication criteria? Is the manuscript technically sound, and do the data support the conclusions? The manuscript must describe methodologically and ethically rigorous research with conclusions that are appropriately drawn based on the data presented.? Is the manuscript technically sound, and do the data support the conclusions? The manuscript must describe methodologically and ethically rigorous research with conclusions that are appropriately drawn based on the data presented.

Reviewer #1: Yes

Reviewer #2: Yes

2. Has the statistical analysis been performed appropriately and rigorously?

Reviewer #1: Yes

Reviewer #2: Yes

3. Have the authors made all data underlying the findings in their manuscript fully available (please refer to the Data Availability Statement at the start of the manuscript PDF file)?

The PLOS Data policy requires authors to make all data underlying the findings described in their manuscript fully available without restriction, with rare exception. The data should be provided as part of the manuscript or its supporting information, or deposited to a public repository. For example, in addition to summary statistics, the data points behind means, medians and variance measures should be available. If there are restrictions on publicly sharing data—e.g. participant privacy or use of data from a third party—those must be specified.requires authors to make all data underlying the findings described in their manuscript fully available without restriction, with rare exception. The data should be provided as part of the manuscript or its supporting information, or deposited to a public repository. For example, in addition to summary statistics, the data points behind means, medians and variance measures should be available. If there are restrictions on publicly sharing data—e.g. participant privacy or use of data from a third party—those must be specified.

Reviewer #1: Yes

Reviewer #2: Yes

4. Is the manuscript presented in an intelligible fashion and written in standard English?

Reviewer #1: Yes

Reviewer #2: Yes

Reviewer #1: This manuscript offers a valuable and thorough analysis on the topic with practical public health implications. However, few minor improvements can be done to improve the overall quality of the study.

1.) Abstract

The concluding statement in the abstract may be strengthened by explicitly stating some key numerical findings of the review.

2.) Introduction

The gap in BMI-specific analysis in previous reviews is properly identified; however, the unique contributions of this review could be highlighted more by more clearly distinguishing it from earlier meta-analyses conducted.

3.) Materials and Methods

Clarify if publication bias was statistically assessed in addition to funnel plots.

Also add brief explanation on how “high risk of bias” studies were defined.

4.) Results

The results section is detailed and clearly presented with subgroup and sensitivity analyses. The statistical analysis is done appropriately.

5.) Discussion

The discussion has some repetition of results. Consider reducing.

6.) Conclusion

The final paragraph of the conclusion section will benefit from explanation of the relevance of findings to their implications for public health policy and clinical guidelines.

7.) Figures

Please consider uploading the figures in a higher resolution for better clarity.

Reviewer #2: Manuscript Title: Impact of yoga on cardiometabolic health in adults with overweight or obesity: A systematic review and meta-analysis of randomized controlled trials

Journal: PLOS Global Public Health

Authors: Wasityastuti et al.

Decision: Minor Revisions

Rationale: Thank you for the opportunity to review this comprehensive and well-conducted systematic review and meta-analysis. The study addresses a clinically relevant topic and employs rigorous methods. Below are my detailed comments, organized by section, to further strengthen the manuscript prior to publication.

Comments

1. Line 54: The search cutoff date is November 2024, indicating that over a year has passed since the searches were performed. Both the Cochrane Collaboration and the Joanna Briggs Institute advise updating literature searches after 12 months, as new relevant studies meeting inclusion criteria may have been published. I recommend updating the search to find and include any recent articles that are relevant.

2. Please include some quantitative results from the meta-analysis in the abstract.t.

3. Lines 126-132: The use of WHO BMI thresholds (≥23 for Asians, ≥25 for others) is appropriate. However, the manuscript states that for studies not explicitly mentioning overweight/obesity, the baseline BMI was extracted from the full text. Please clarify the process if only the mean group BMI met the threshold, but individual participant data was not available. Was the study included if the mean BMI of the intervention/control group met the threshold, even if some participants individually did not? This has implications for the specificity of the findings to an overweight/obese population.

4. Lines 126-132: The exclusion of participants with comorbidities (T2DM, CAD, etc.) is justified to isolate the effect of yoga. However, this significantly limits the clinical applicability of the findings, as overweight/obesity often coexists with these conditions. This should be highlighted as a key limitation affecting generalizability to the broader target population in clinical practice (see also Lines 564-566).

5. The definition of "yoga" as a complex intervention is clear. However, for the "combined intervention" category (yoga plus other health interventions), the protocol states the single or complementary effect of yoga must be examinable. Please specify in the methods how this was determined from the study design (e.g., only included if there was a yoga-only arm vs. a control arm, allowing isolation of yoga's effect).

6. The comparator includes "other types of physical activity interventions." Consider adding a brief rationale for this choice (e.g., to assess yoga's efficacy relative to other common exercise prescriptions). This strengthens the review's aim to inform exercise selection.

7. The vote-counting approach for narrative synthesis is clearly described. The use of WebPlotDigitizer for data extraction from graphs is noted, which is a common practice. Consider adding a sentence on how data extracted this way was verified (e.g., by a second reviewer) to ensure accuracy.

8. The rationale for subgroup analyses (ethnicity, dose) is well-founded. However, the protocol mentions dose–response effects (Lines 256-260) but the primary subgroup analyses in the results (Lines 389-406) focus on categorical cut-offs (e.g., ≥12 weeks, ≥60 min/session). Consider if a meta-regression analyzing dose (total minutes of yoga) as a continuous variable was explored and, if not, mention it as a limitation for future research. This could provide a more nuanced dose-response relationship.

9. In the "Quality of evidence" section (Lines 419-447), it is stated that for most outcomes the results did not surpass the MCID. This is a critical point for clinical interpretation. Consider emphasizing this more prominently in the Abstract (Lines 63-65) and Discussion to ensure readers appreciate that while statistically significant, changes in lipid profiles, for example, were modest in clinical magnitude.

10. Lines 60-61: The statement "While 25 studies did not specifically recruit individuals with obesity, the average BMI... fell within the overweight or obese range" is important but slightly ambiguous. Rephrase for clarity: e.g., "Although most studies (25/28) did not explicitly recruit an overweight/obese population, the mean baseline BMI of participants met our inclusion criteria for overweight/obesity."

11. Lines 60-61: The statement "While 25 studies did not specifically recruit individuals with obesity, the average BMI... fell within the overweight or obese range" is important but slightly ambiguous. Rephrase for clarity: e.g., "Although most studies (25/28) did not explicitly recruit an overweight/obese population, the mean baseline BMI of participants met our inclusion criteria for overweight/obesity."

12. Lines 60-61: The statement "While 25 studies did not specifically recruit individuals with obesity, the average BMI... fell within the overweight or obese range" is important but slightly ambiguous. Rephrase for clarity: e.g., "Although most studies (25/28) did not explicitly recruit an overweight/obese population, the mean baseline BMI of participants met our inclusion criteria for overweight/obesity."

13. Lines 60-61: The statement "While 25 studies did not specifically recruit individuals with obesity, the average BMI... fell within the overweight or obese range" is important but slightly ambiguous. Rephrase for clarity: e.g., "Although most studies (25/28) did not explicitly recruit an overweight/obese population, the mean baseline BMI of participants met our inclusion criteria for overweight/obesity."

14. Lines 60-61: The statement "While 25 studies did not specifically recruit individuals with obesity, the average BMI... fell within the overweight or obese range" is important but slightly ambiguous. Rephrase for clarity: e.g., "Although most studies (25/28) did not explicitly recruit an overweight/obese population, the mean baseline BMI of participants met our inclusion criteria for overweight/obesity."

**Do you want your identity to be public for this peer review?** For information about this choice, including consent withdrawal, please see our Privacy Policy..

Reviewer #1: No

Reviewer #2: **Yes:** Mohammad AlghosiMohammad AlghosiMohammad AlghosiMohammad Alghosi

---

## [Decision Letter · Decision Letter 1]

10 Mar 2026

Impact of yoga on cardiometabolic health in adults with overweight or obesity: A systematic review and meta-analysis of randomized controlled trials

PGPH-D-25-01226R1

Dear Dr. Wibowo,

We are pleased to inform you that your manuscript 'Impact of yoga on cardiometabolic health in adults with overweight or obesity: A systematic review and meta-analysis of randomized controlled trials' has been provisionally accepted for publication in PLOS Global Public Health.

Best regards,

Zulkarnain Jaafar

Academic Editor

Reviewer Comments (if any, and for reference):

Reviewer's Responses to Questions

**Comments to the Author**

Reviewer #2: All comments have been addressed

publication criteria? Is the manuscript technically sound, and do the data support the conclusions? The manuscript must describe methodologically and ethically rigorous research with conclusions that are appropriately drawn based on the data presented.? Is the manuscript technically sound, and do the data support the conclusions? The manuscript must describe methodologically and ethically rigorous research with conclusions that are appropriately drawn based on the data presented.

Reviewer #2: Yes

3. Has the statistical analysis been performed appropriately and rigorously?

Reviewer #2: Yes

4. Have the authors made all data underlying the findings in their manuscript fully available (please refer to the Data Availability Statement at the start of the manuscript PDF file)?

The PLOS Data policy requires authors to make all data underlying the findings described in their manuscript fully available without restriction, with rare exception. The data should be provided as part of the manuscript or its supporting information, or deposited to a public repository. For example, in addition to summary statistics, the data points behind means, medians and variance measures should be available. If there are restrictions on publicly sharing data—e.g. participant privacy or use of data from a third party—those must be specified.requires authors to make all data underlying the findings described in their manuscript fully available without restriction, with rare exception. The data should be provided as part of the manuscript or its supporting information, or deposited to a public repository. For example, in addition to summary statistics, the data points behind means, medians and variance measures should be available. If there are restrictions on publicly sharing data—e.g. participant privacy or use of data from a third party—those must be specified.

Reviewer #2: Yes

5. Is the manuscript presented in an intelligible fashion and written in standard English?

Reviewer #2: Yes

**Reviewer #2**: I really appreciated the study. Congratulations to the authors.: I really appreciated the study. Congratulations to the authors.: I really appreciated the study. Congratulations to the authors.: I really appreciated the study. Congratulations to the authors.

**Do you want your identity to be public for this peer review?** For information about this choice, including consent withdrawal, please see our Privacy Policy..

Reviewer #2: **Yes:** Mohammad AlghosiMohammad AlghosiMohammad AlghosiMohammad Alghosi
